# Beyond Static Retrieval: Opportunities and Pitfalls of Iterative Retrieval in GraphRAG

## Abstract

Retrieval-augmented generation (RAG) is a powerful paradigm for improving large language models (LLMs) on knowledge-intensive question answering. Graph-based RAG (GraphRAG) leverages entity–relation graphs to support multi-hop reasoning, but most systems still rely on static retrieval. When crucial evidence, especially bridge documents that connect disjoint entities, is absent, reasoning collapses and hallucinations persist. Iterative retrieval, which performs multiple rounds of evidence selection, has emerged as a promising alternative, yet its role within GraphRAG remains poorly understood. We present the first systematic study of iterative retrieval in GraphRAG, analyzing how different strategies interact with graph-based backbones and under what conditions they succeed or fail. Our findings reveal clear opportunities: iteration improves complex multi-hop questions, helps promote bridge documents into leading ranks, and different strategies offer complementary strengths. At the same time, pitfalls remain: naive expansion often introduces noise that reduces precision, gains are limited on single-hop or simple comparison questions, and several bridge evidences still be buried too deep to be effectively used. Together, these results highlight a central bottleneck, namely that GraphRAG's effectiveness depends not only on recall but also on whether bridge evidence is consistently promoted into leading positions where it can support reasoning chains. To address this challenge, we propose *Bridge-Guided Dual-Thought-based Retrieval (BDTR)*, a simple yet effective framework that generates complementary thoughts and leverages reasoning chains to recalibrate rankings and bring bridge evidence into leading positions. BDTR achieves consistent improvements across diverse GraphRAG settings and provides guidance for the design of future GraphRAG systems.

## 1 Introduction

In recent years, retrieval-augmented generation (RAG) has become a core paradigm for enhancing large language models (LLMs) on knowledge-intensive question answering (Xia et al., 2024; Lewis et al., 2020; Gao et al., 2023; Fan et al., 2024). By grounding generation in external evidence, RAG is able to effectively mitigate hallucination (Ayala & Béchard, 2024; Niu et al., 2024). Despite its tremendous success, standard RAG systems often struggle with multi-hop reasoning, where multiple pieces of evidence are required to be linked across retrieval and inference (Tang & Yang, 2024; Saleh et al., 2024; Han et al., 2025).

To address this challenge, graph-based retrieval-augmented generation (GraphRAG) has emerged as a promising extension (Edge et al., 2024; Han et al., 2024; He et al., 2024; Wu et al., 2025; Jiang et al., 2024). By integrating entity–relation knowledge graphs into the retrieval pipeline, GraphRAG supports structured reasoning over multi-hop paths and has achieved strong performance on multi-hop QA tasks (Zou et al., 2025; Jimenez Gutierrez et al., 2024; Gutiérrez et al., 2025; Mavromatis & Karypis, 2025). However, most existing GraphRAG systems rely on single-shot static retrieval. If crucial evidence is absent from the selected candidates, the reasoning process collapses, and hallucinations persist (Luo et al., 2025a; Guo et al., 2025).

Meanwhile, iterative retrieval has gained growing attention in the broader RAG literature (Trivedi et al., 2022a; Jiang et al., 2025; Lee et al., 2025). Instead of committing to one retrieval step, iterative methods allow models to perform multiple retrieval rounds during reasoning, progres-

sively refining or expanding the evidence set (Shao et al., 2023). This dynamic process can improve coverage, reduce hallucination, and has shown benefits in chain-of-thought and self-ask style frameworks (Trivedi et al., 2022a). Recent systems such as GFM-RAG (Luo et al., 2025b) and HippoRAG (Jimenez Gutierrez et al., 2024) include limited explorations using iterative retrieval. However, their analyses remain unsystematic, leaving open a fundamental question:

> **Can iterative retrieval reliably improve GraphRAG, and under what conditions does it succeed or fail?**

In this work, we present the first comprehensive study of iterative retrieval in GraphRAG. We integrate four representative GraphRAG backbones with four iterative retrieval strategies and evaluate them systematically across multi-hop QA benchmarks. Our analysis uncovers both *opportunities* and *pitfalls*:

- **Opportunities.** (1) Iterative retrieval substantially improves complex multi-hop questions, especially those requiring *bridge documents*—intermediate facts that connect otherwise disjoint entities. (2) Different iterative strategies exhibit complementary strengths, indicating potential for combination. (3) Iteration can act as an implicit re-ranking mechanism: by repeatedly updating scores, gold documents are progressively promoted into the leading positions, which leads to a sharp improvement in recall in the top ranks.
- **Pitfalls.** (1) Simply expanding the number of retrieved documents is not always beneficial: while recall may increase, the additional noise often dilutes precision and undermines QA accuracy. (2) For single-hop or simple comparison questions, iterative retrieval offers little to no benefit, and may even harm performance. (3) Even when gold bridge documents are retrieved, many remain buried beyond the leading positions, making them effectively unusable for reasoning.

Together, these findings highlight a central **bottleneck**: GraphRAG's success depends not only on overall recall coverage, but on whether bridge-bearing evidence is consistently promoted into the leading positions where it can be used to complete reasoning chains. This perspective explains our observations: performance improves significantly on bridge-type questions once the required intermediate evidence is made available in the leading ranks.

Building on this insight, we propose *Bridge-Guided Dual-Thought-based Retrieval (BDTR)*, a simple yet effective iterative framework that explicitly targets this bottleneck. BDTR generates dual thoughts at each reasoning step to broaden coverage with complementary retrieval prompt, and leverages reasoning chains to recalibrate rankings and promote bridge evidence into the leading positions. This design consistently improves multi-hop QA across diverse GraphRAG backbones and datasets, offering practical guidance for future GraphRAG systems. Our contributions are threefold:

- We introduce the first systematic study of iterative retrieval in GraphRAG, covering multiple models and strategies.
- We provide new empirical findings that reveal both the strengths and weaknesses of iterative retrieval, identifying the bridge bottleneck as the decisive factor.
- We propose BDTR, a reasoning chain-guided framework that addresses this bottleneck and achieves consistent gains across backbones and datasets.

## 2    PRELIMINARY STUDIES

To probe the effectiveness of iterative retrieval in GraphRAG, we begin by asking a question: *does iterative retrieval indeed improve performance, especially on multi-hop questions where GraphRAG is expected to shine?* To answer this, we conduct experiments across multiple datasets to establish the overall effectiveness. However, rather than stopping at the observation that iterative retrieval works, we seek to unpack the underlying mechanisms. Specifically, we investigate: (i) Which question types benefit most, and which do not? (ii) From a retrieval perspective, how does recall explain these improvements? (iii) How many rounds are necessary before the benefits saturate? (iv) Do different iterative strategies exhibit distinct behaviors?

These guiding questions structure our preliminary studies. By addressing them, we not only validate the effectiveness of iterative retrieval for GraphRAG, but also develop deeper intuitions that motivate the principled framework proposed in this work.

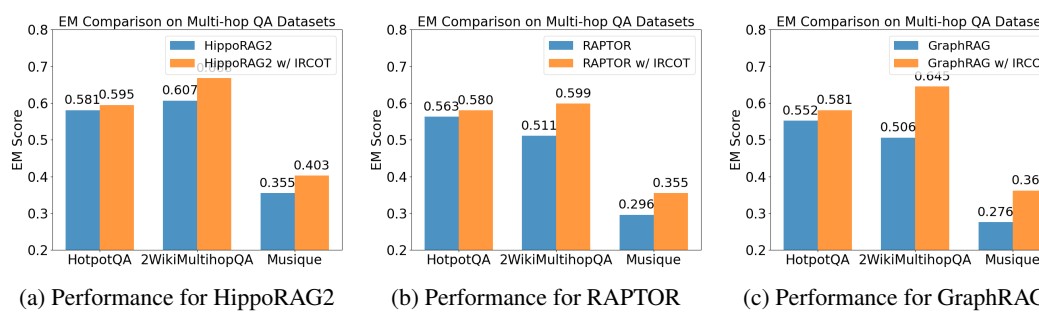

Figure 1: EM Comparison on Multi-hop QA Datasets.

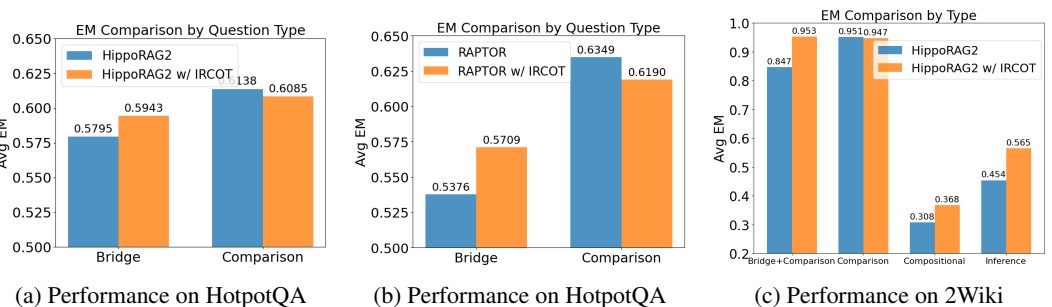

Figure 2: EM Comparison by Question Type.

## 2.1 EXPERIMENTAL SETTINGS

We begin by establishing the experimental setup, specifying the backbone model, iterative module, and evaluation datasets, so that subsequent analyses can be interpreted under a consistent framework. Specifically, we adopt HippoRAG2, RAPTOR, and GraphRAG as backbone models, and integrate the iterative method IRCOT (Trivedi et al., 2022a). Experiments are carried out on three widely used multi-hop QA datasets: HotpotQA, 2WikiMultiHopQA, and MuSiQue. Following prior work (Gutiérrez et al., 2025), we use Exact Match (EM) and F1 as the evaluation metrics. This setup allows us to isolate the effect of iterative retrieval while controlling for other modeling factors.

## 2.2 OVERALL EFFECTIVENESS

We first verify whether iterative retrieval yields consistent gains in overall accuracy across datasets, providing a high-level validation of its effectiveness before delving into finer-grained analyses. Figure 1 shows that incorporating IRCOT consistently improves HippoRAG2, RAPTOR, GraphRAG across all three datasets. On HotpotQA, the gains are modest, but on 2WikiMultiHopQA and MuSiQue the improvements are more substantial. These results confirm that iterative retrieval is particularly valuable in settings that demand complex reasoning chains, as additional retrieval rounds help surface supporting evidence that static retrieval often overlooks. This suggests that the current design of GraphRAG underutilizes its potential: while the graph structure provides a powerful foundation, its effectiveness is limited by the quality of retrieved evidence for multi-hop QA.

## 2.3 QUESTION-TYPE ANALYSIS

Beyond aggregate performance, it is crucial to identify which types of questions benefit most from iterative retrieval, since multi-hop reasoning demands vary across query categories. We therefore break down the results by question type to reveal when iterative retrieval is most beneficial. HotpotQA includes two types of questions: Bridge and Comparison. 2WikiMultiHopQA contains four types: Bridge+Comparison, Comparison, Compositional, and Inference.

From the Figure 2, a clear pattern emerges: iterative retrieval yields the greatest benefit on *Bridge* and other multi-hop questions that require linking disparate pieces of evidence. These questions

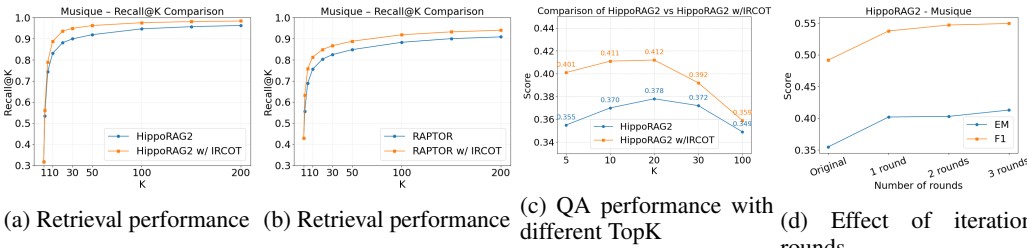

(a) Retrieval performance  (b) Retrieval performance  (c) QA performance with different TopK  (d) Effect of iteration rounds

Figure 4: EM Comparison on Multi-hop QA Datasets.

demand identification of intermediate bridge entities that are rarely stated explicitly in the query, making them particularly challenging for static retrieval. In contrast, iterative retrieval offers little to no improvement on simple Comparison questions, and can even lead to slight performance drops due to over-retrieval and noise accumulation. Despite being recognized for its strength in multi-hop reasoning, GraphRAG struggles to handle more complex multi-hop questions without iterative retrieval. We discribe a example in Fig 7 in Appendix.

## 2.4 Understanding Improvements through Recall

The preceding analysis of question types (Section 2.3) shows that iterative retrieval brings the largest gains on *Bridge*-style questions, where intermediate entities must be identified to connect disjoint pieces of evidence. This already suggests that the core challenge lies in whether such bridge-bearing documents can be surfaced. To further clarify this mechanism, we now examine retrieval performance through Recall@K.

High overall coverage (i.e., whether gold documents can eventually be retrieved at large $K$) is a necessary condition, but it is not sufficient for effective reasoning in multi-hop QA. What truly matters is whether the critical bridge documents appear within the leading positions that the model is most likely to use. In other words, once coverage is ensured, improving recall at these leading positions (e.g., top-5 or top-10) becomes critical.

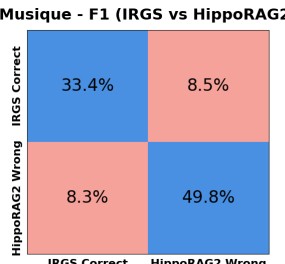

Figure 3: Complementary.

As shown in Fig. 4a, baseline HippoRAG2 already achieves high coverage at large $K$ (Recall@100 $\approx$ 0.95), yet many gold documents are absent from the leading positions (such as Recall@5 or Recall@10), leaving them effectively unused. Iterative retrieval mitigates this gap by boosting recall at small-$K$ ranges (e.g., Recall@5 increases from 0.7435 to 0.7894, Recall@10 from 0.8309 to 0.8879). This improvement in recall directly corresponds to the performance gains on bridge-type questions highlighted in Section 2.3. However, a substantial gap remains between recall at top-10 and top-200, particularly for RAPTOR, as shown in Fig 4b. This indicates that even with iterative retrieval, a large portion of gold documents—especially critical bridge evidence—are still buried deep in the ranked list. At the same time, Fig. 4c shows that simply enlarging $K$ does not guarantee improvements: when $K$ becomes too large, irrelevant documents accumulate and introduce noise. Thus, the benefit of iterative retrieval lies not in broadening the pool, but in selectively elevating the bridge-bearing evidence into the leading positions where it can directly support reasoning.

In summary, GraphRAG's bottleneck is not merely coverage, but whether the *bridge documents* are ranked high enough to be used. Performance improves most when the retrieval process both raises these documents to the leading positions and correctly identifies them as the links that complete the reasoning chain.

## 2.5 Impact of the Number of Rounds

An important design choice in iterative retrieval is how many rounds to perform. On the one hand, additional rounds may surface evidence that is missed initially; on the other hand, excessive rounds risk introducing redundancy or irrelevant documents. We therefore examine how performance evolves as the number of rounds increases.

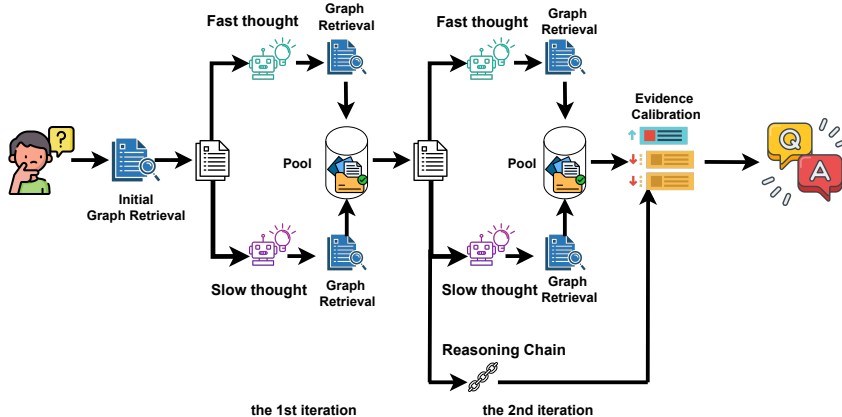

Figure 5: Illustration of our framework BDTR, shown here with two iterations as an example. In each reasoning step, the model generates two thoughts to drive retrieval and constructs a reasoning chain that encodes intermediate bridge cues. The retrieved documents from the two thoughts provide diverse and complementary evidence, while the bridge-guided calibration module adjusts their ranking to ensure that critical bridge facts appear in leading position for reasoning.

As shown in Fig. 4d, moving from a single round to two rounds of IRCOT yields substantial improvements, demonstrating that two additional pass is often sufficient to retrieve the missing bridge evidence. However, extending to three or more rounds provides only diminishing returns, suggesting that two iterations achieve the most favorable cost–benefit balance and nearly reach convergence.

## 2.6 COMPLEMENTARITY OF ITERATIVE METHODS

Beyond the number of rounds, another question is whether different iterative strategies capture distinct aspects of the evidence space. In that case, combining them could extend coverage and improve robustness. To investigate this, we compare two representative strategies: IRCOT (Trivedi et al., 2022a) and IRGS (Shao et al., 2023).

As illustrated in Figure 3, each method succeeds on a different subset of questions, indicating that they uncover complementary evidence. This complementarity suggests that no single iterative method is universally optimal. Instead, combining strategies—or adaptively selecting among them depending on the query—has the potential to achieve broader coverage and more reliable performance than any individual approach.

## 2.7 SUMMARY OF OBSERVATIONS

Our preliminary studies reveal both the opportunities and the challenges of applying iterative retrieval in GraphRAG.

**Opportunities. 1)** Iterative retrieval consistently enhances performance on complex multi-hop questions, particularly Bridge-type questions that require identifying implicit intermediate entities. **2)** The performance gains can be attributed to the reranking effect of iterative retrieval, which improves recall at small cutoffs (e.g., Recall@5, Recall@10) and thereby increases the likelihood that critical facts are utilized in reasoning. **3)** Different iterative strategies (e.g., IRCOT vs. IRGS) demonstrate complementary strengths, suggesting that combining or adaptively selecting among them could further improve coverage and robustness.

**Pitfalls. 1)** Iterative retrieval offers little to no benefit on simple Comparison questions, and in some cases even reduces performance due to over-thinking. **2)** Increasing the number of rounds beyond two generally leads to diminishing returns, reflecting an efficiency–effectiveness trade-off where additional complexity brings little incremental benefit. **3)** Although most gold documents—including bridge documents—can be retrieved, many fail to appear within the leading positions, thereby limiting their practical usefulness for reasoning.

## 3 METHOD

Based on our analysis, we design a new method guided by two key insights: **(1) Opportunity:** Different iterative methods exhibit complementary strengths. **(2) Pitfall:** While gold documents—particularly those containing bridge facts—can be retrieved, not all are ranked in the leading positions necessary for effective reasoning.

These insights motivate two core components. First, instead of relying on a single reasoning path, each step produces two thoughts with complementary. This design broadens coverage by combining distinct retrieval signals, ensuring that more gold evidence is captured. Second, we introduce a bridge-aware reranking mechanism that uses cues from the evolving reasoning chain to elevate bridge-bearing documents into the top ranks, where they can be effectively used. Together, these components improve both coverage and ranking quality, addressing the main bottlenecks identified in our preliminary analysis. The concrete algorithm is shown in Appendix A.3.

An overview of our framework, **Bridge-Guided Dual-Thought-based Retrieval (BDTR)**, is shown in Figure 5. BDTR consists of two functional modules: **Dual-Thought-based Retrieval (DTR)** expands coverage by generating two retrieval signals per step, capturing evidence that a single strategy might miss. **Bridge-Guided Evidence Calibration (BGEC)** improves usability by promoting documents likely to contain bridge facts into the top ranks. These two modules operate jointly, with DTR ensuring breadth of coverage and BGEC ensuring that the most relevant evidence is prioritized.

### 3.1 DUAL-THOUGHT-BASED RETRIEVAL (DTR)

Our analysis indicates that different iterative strategies capture complementary evidence. To exploit this property, each reasoning step generates two thoughts: one biased toward direct-answering passages and the other toward bridge-seeking relations. Each thought is issued as an independent query to the backbone graph retriever.

**Initialization.** Let $Q$ be the original question and $f_{\text{ret}}$ be the backbone retriever. We use $d \in \mathcal{D}$ to denote a candidate document $d$ from a corpus $\mathcal{D}$. The first round retrieves a set of documents:

$$D_0 = f_{\text{ret}}(Q), \tag{1}$$

where each $d \in D_0$ is returned with a retrieval score $\hat{s}(d \mid Q)$. The scores are generated by GraphRAG backbones. These documents and scores form the initial pool $P_0$.

**Iterative dual-thought retrieval.** At iteration $t \geq 1$, two complementary thoughts are generated, each conditioned on the retrieved pool from the previous step, with $P_0$ used for the first iteration and $P_{t-1}$ for subsequent ones. We denote them by $q_t^{\text{FT}}$ and $q_t^{\text{ST}}$. Specifically, $q_t^{\text{FT}}$ corresponds to the **fast thought** (FT), whereas $q_t^{\text{ST}}$ corresponds to the **slow thought** (ST), as illustrated in Fig. 8 in the Appendix. These thoughts are produced using distinct prompts to capture complementary perspectives, and are independently submitted to the retriever, yielding two sets of documents:

$$D_t^{\text{FT}} = f_{\text{ret}}(q_t^{\text{FT}}), \qquad D_t^{\text{ST}} = f_{\text{ret}}(q_t^{\text{ST}}). \tag{2}$$

**Pool update and sorting.** The candidate pool is expanded as

$$P_t = P_{t-1} \cup D_t^{\text{FT}} \cup D_t^{\text{ST}}, \tag{3}$$

and the score of each document is updated by

$$s_t(d) \leftarrow \max\Big(s_{t-1}(d), \ \hat{s}(d \mid q_t^{\text{FT}}), \ \hat{s}(d \mid q_t^{\text{ST}})\Big), \quad \forall d \in P_t, \tag{4}$$

where $s_t(d)$ is the updated score of document $d$ at iteration $t$ and $\hat{s}(d \mid q)$ is the score assigned by the retriever to document $d$ given query $q$. By taking the maximum in equation 4, each document preserves is able to preserve it's highest score. Afterward, all documents in $P_t$ are **re-sorted** according to the updated scores.

In practice, combining the two trajectories enlarges the evidence frontier across iterations. By repeating this process, DTR improves the likelihood that gold documents are covered and retained with strong scores, paving the way for subsequent ranking calibration.

## 3.2 BRIDGE-GUIDED EVIDENCE CALIBRATION (BGEC)

Answering complex questions typically relies on bridge documents that provide the necessary connections between facts. However, such evidence is hard to surface, as it may not be explicitly reflected in the query. Furthermore, even if it is retrieved, it often appears far down in the ranking where it cannot be used. Without these bridge documents, the reasoning chain breaks, making it impossible to answer the question even if other supporting facts are present. This motivates a bridge-guided calibration step that explicitly identifies and promotes such overlooked documents.

**Bridge-aware selection.** In the final iteration, besides generating dual thoughts, the model also produces a reasoning chain $RC$ that encodes potential bridging cues. We denote the pool of candidate documents after the last iteration as $P_R$. The pool $P_R$ together with $RC$ is sent to an LLM verifier. The verifier does not assign scores but directly selects documents that align with the reasoning chain:

$$\mathcal{G} = \{d \in P_R \mid \text{Verifier}(d, RC) = 1\}, \tag{5}$$

where $\text{Verifier}(d, RC) = 1$ indicates that the document $d$ is judged to support $RC$. The concrete prompt of verifier is shown in Fig 9 in Appendix. The selected documents $\mathcal{G}$ are then promoted to the top of $P_R$, ensuring that bridge-supporting evidence is made accessible to reasoning.

**Final selection.** After calibration, we further filter the pool to produce a compact and reliable context for QA. Let $P_{50}$ denote the top-50 documents in $P_R$ after re-ranking, and let $\mu$ and $\sigma$ be the mean and standard deviation of their scores. We define the final set as

$$\mathcal{D}_{\text{final}} = \{d \in P_R \mid s(d) \geq \mu + \sigma\}, \tag{6}$$

with the safeguard that at least 5 documents are always retained. This criterion guaranteeing that enough strong evidence remains to support the reasoning chain.

**Effect.** BGEC therefore leverages reasoning chains to uncover bridge documents that ordinary retrieval misses or under-ranks, promotes them to the top positions, and applies a statistically robust cutoff to refine the final evidence set. This step significantly improves the usability of retrieved evidence by closing gaps in the evidence chain.

## 4 EXPERIMENTS

In our experiments, we aim to answer the following research questions: **RQ1:** How effective is the proposed BDTR framework when applied to state-of-the-art GraphRAG backbones for multi-hop QA? **RQ2:** How does BDTR compare with other iterative retrieval approaches? **RQ3:** What is the impact of the two core modules, DTR and BGEC, on overall performance?

### 4.1 EXPERIMENTAL SETTINGS

**Datasets.** We evaluate our approach on three widely used multi-hop QA benchmarks: HotpotQA (Yang et al., 2018), 2WikiMultiHopQA, and MuSiQue (Trivedi et al., 2022b). Following HippoRAG (Jimenez Gutierrez et al., 2024), we randomly sample 1,000 queries from each dataset. Since all of these benchmarks require reasoning across multiple evidence documents, they provide an appropriate setting for assessing the effectiveness of iterative retrieval strategies. In addition, we evaluate on a single-hop dataset to examine the effectiveness of both the baselines and our method. Specifically, we sample 1,000 queries from PopQA (Mallen et al., 2022), using the corpus constructed from the December 2021 Wikipedia dump as in HippoRAG (Jimenez Gutierrez et al., 2024).

**GraphRAG Backbones.** We integrate our method with four representative GraphRAG variants: HippoRAG2 (Gutiérrez et al., 2025) (PPR-based), RAPTOR (Sarthi et al., 2024) (tree-based), GFM-RAG (Luo et al., 2025b) (GNN-based), and GraphRAG (Edge et al., 2024) (community-based), covering diverse retrieval paradigms.

**Iterative Baselines.** We compare BDTR against state-of-the-art iterative methods, including IRCOT (Trivedi et al., 2022a), IRGS (Shao et al., 2023), TOG (Sun et al., 2023),

| Framework | Method | HotpotQA | | 2WikiMultiHopQA | | MuSiQue | |
|---|---|---|---|---|---|---|---|
| | | EM | F1 | EM | F1 | EM | F1 |
| HippoRAG2 | Original | 0.581 | 0.7372 | 0.607 | 0.7059 | 0.355 | 0.4917 |
| | IRCOT | 0.595 | 0.7493 | **0.668** | **0.7683** | 0.403 | 0.5469 |
| | GCOT | 0.597 | 0.7491 | 0.662 | 0.7652 | 0.410 | 0.5417 |
| | TOG | 0.590 | 0.7381 | 0.630 | 0.7330 | 0.374 | 0.5352 |
| | IRGS | 0.593 | 0.7484 | 0.652 | 0.7546 | 0.404 | 0.5436 |
| | **Our** | **0.607** | **0.7590** | 0.664 | 0.7651 | **0.423** | **0.5613** |
| RAPTOR | Original | 0.563 | 0.7080 | 0.511 | 0.5726 | 0.296 | 0.4179 |
| | IRCOT | 0.580 | 0.7266 | 0.599 | 0.6870 | 0.355 | 0.4905 |
| | GCOT | 0.588 | 0.7261 | 0.586 | 0.6814 | 0.358 | 0.4882 |
| | TOG | 0.560 | 0.7041 | 0.535 | 0.6062 | 0.297 | 0.4229 |
| | IRGS | 0.581 | 0.7262 | 0.592 | 0.6833 | 0.352 | 0.4779 |
| | **Our** | **0.598** | 0.7444 | 0.665 | 0.7608 | **0.399** | **0.5400** |
| GFM-RAG | Original | 0.546 | 0.6820 | 0.672 | 0.7546 | 0.279 | 0.3982 |
| | IRCOT | 0.562 | 0.7066 | 0.697 | 0.7768 | 0.320 | 0.4440 |
| | GCOT | 0.556 | 0.6924 | 0.690 | 0.7711 | 0.329 | 0.4484 |
| | TOG | 0.558 | 0.7029 | 0.697 | 0.7847 | 0.321 | 0.4369 |
| | IRGS | 0.560 | 0.7184 | 0.693 | 0.7745 | 0.307 | 0.4380 |
| | **Our** | 0.585 | **0.7462** | **0.726** | **0.8046** | 0.370 | 0.4943 |
| GraphRAG | Original | 0.552 | 0.6983 | 0.506 | 0.5757 | 0.276 | 0.4017 |
| | IRCOT | 0.581 | 0.7283 | 0.645 | 0.7533 | 0.362 | 0.5060 |
| | GCOT | 0.560 | 0.7041 | 0.652 | 0.7582 | 0.349 | 0.4917 |
| | TOG | 0.561 | 0.7080 | 0.541 | 0.6196 | 0.303 | 0.4309 |
| | IRGS | 0.566 | 0.7126 | 0.580 | 0.6842 | 0.324 | 0.4609 |
| | **Our** | 0.595 | 0.7459 | 0.655 | 0.7471 | 0.386 | 0.5321 |
| **Ave. Improvement** | | **2.47%** | **2.51%** | **3.74%** | **2.85%** | **8.41%** | **6.73%** |

Table 1: EM and F1 performance across multi-hop QA datasets. Each framework is evaluated with different iterative retrieval methods. Highlighted are the results ranked first and second.

and GCOT (Jin et al., 2024), to test whether our bridge-guided design offers advantages beyond existing strategies. The corresponding prompts are illustrated in figs. 10 to 13.

**Implementation and Evaluation Metrics.** For implementation, we adopt the official codebases of HippoRAG2 and GFM-RAG, and reimplement RAPTOR and GraphRAG within the HippoRAG2 framework. Across all methods, GPT-4o-mini is used as the iterative reasoning engine, the verifier, and the generator for producing answers. We report two widely used metrics: Exact Match (EM) and F1. EM measures the percentage of predictions that exactly match the ground-truth answers, providing a strict indicator of correctness. F1 measures the token-level overlap between the predicted and gold answers, offering a softer evaluation that captures partial correctness. Together, EM and F1 provide a balanced view of QA performance in terms of both precision and recall. For all experiments, the number of iteration is set to 2.

| Methods | EM | F1 |
|---|---|---|
| Original | 0.419 | 0.5603 |
| +IRCOT | 0.425 | 0.5629 |
| +GCOT | 0.429 | 0.5662 |
| +TOG | 0.419 | 0.5615 |
| +IRGS | 0.426 | 0.5617 |
| +Our | **0.435** | **0.5735** |

Table 2: Results on Single-Hop dataset PopQA with HippoRAG2.

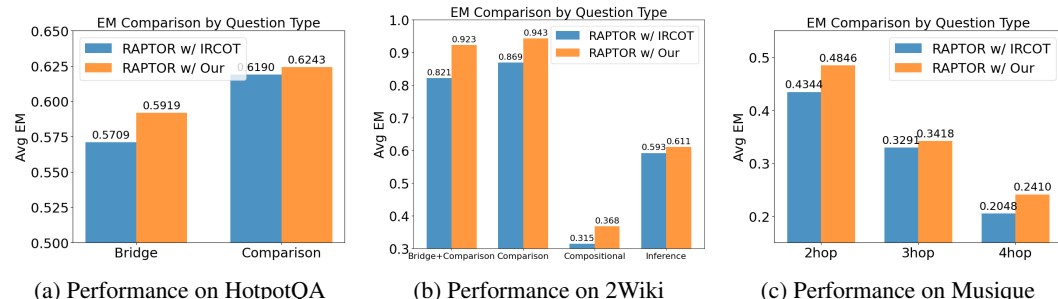

Figure 6: EM Comparison on Multi-hop QA Datasets with Different Question Type.

## 4.2 MAIN RESULTS

In this section, we evaluate the performance of our proposed BDTR framework with various GraphRAG backbones and compare it against iterative retrieval baselines. The **efficiency analysis** are shown in Appendix A.4. The **ablation study** is illustrated in Appendix A.5

**RQ1: QA Performance Comparison.** We apply BDTR to different GraphRAG variants, including HippoRAG2, RAPTOR, GFM-RAG, and GraphRAG. The results in Table 1 show that BDTR consistently improves performance across all backbones. On average, our method achieves an improvement of 11.0% in EM and 8.50% in F1 over HippoRAG2, 23.72% in EM and 22.41% in F1 over RAPTOR, and 15.94% in EM and 13.39% in F1 over GFM-RAG across three multi-hop QA datasets. These consistent gains demonstrate that BDTR is broadly effective across different retrieval paradigms, reinforcing its adaptability to diverse reasoning strategies in multi-hop QA.

**RQ2: Comparison with other iterative methods.** We further compare BDTR with other state-of-the-art iterative retrieval strategies, including IRCOT, IRGS, TOG, and GCOT. As shown in Table 1, BDTR consistently outperforms all baselines across datasets and backbones. For instance, compared to baselines, our method achieves an average improvement of 2.47% in EM and 2.51% on HotpotQA. Against baselines, BDTR improves by 3.74% in EM and 2.85% on 2Wikimultihopqa and BDTR improves by 8.41% in EM and 6.73% on Musique. These results highlight that BDTR not only inherits the benefits of iterative retrieval but also addresses its limitations through bridge-guided calibration, yielding more robust and reliable improvements.

In addition, we analyze performance across different question types. HotpotQA includes two types of questions: Bridge and Comparison. 2WikiMultiHopQA contains four types: Bridge+Comparison, Comparison, Compositional, and Inference. MuSiQue consists of three types: 2-hop, 3-hop, and 4-hop. As shown in Figure 6, our method yields the largest gains on bridge-type questions and delivers consistent improvements on a range of multi-hop questions. Moreover, it alleviates the failure of standard iterative retrieval on comparison questions. Furthermore, we present the retrieval performance on the Musique dataset in Table 3 in Appendix. The results show that our method achieves higher Recall@5 and Recall@10, demonstrating its ability to rank gold documents in the leading positions.

**Performance on single-hop dataset.** In addition to multi-hop datasets, we also evaluated on a single-hop dataset. The results in Table 2 show that, unlike in the multi-hop setting where iterative retrieval often yields significant improvements, these methods provide little to no benefit on single-hop questions. This finding further confirms our earlier observation that iterative retrieval is particularly suited for supporting multi-hop reasoning. On this single-hop dataset, our method achieves improvement over the baselines.

## 5 CONCLUSION

In this work, we presented the first systematic study of iterative retrieval in GraphRAG. Our analysis shows that iteration can promote bridge evidence and improve multi-hop reasoning, but it still leaves some gold documents buried too deep to be effectively used. To overcome this bottleneck, we proposed *Bridge-Guided Dual-Thought-based Retrieval (BDTR)*, which consistently improves performance and provides guidance for future GraphRAG systems.

## 6 ETHICS STATEMENT

I acknowledge that I and all co-authors of this work have read and commit to adhering to the ICLR Code of Ethics. Following HippoRAG (Jimenez Gutierrez et al., 2024), we adopt the same publicly available datasets in our study. We believe this work raises no direct ethical risks beyond standard concerns associated with machine learning research.

## 7 REPRODUCIBILITY STATEMENT

To ensure reproducibility, we detail the experimental settings in Section 2.1 and Section 4.1, present the algorithm in Section A.3, and provide the prompts in Section A.6.

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

# A  APPENDIX

## A.1  USAGE OF LARGE LANGUAGE MODELS

We employ large language models (LLM) to enhance the clarity of our writing and use an LLM-based coding assistant to support code development. In addition, we utilize the LLM as the iterative reasoning engine, the verifier, and the answer generator in our experiments. All outputs produced by large language models have been carefully reviewed, and the authors bear responsibility for the correctness of the final content.

## A.2  RELATED WORK

**GraphRAG.** Graph-based retrieval-augmented generation (GraphRAG) has emerged as a promising paradigm for improving large language models on multi-hop question answering by grounding reasoning in structured entity–relation graphs. A variety of methods have been proposed to integrate graph structure into retrieval. HippoRAG2 (Gutiérrez et al., 2025) adopts personalized PageRank (PPR) to expand retrieval around entity mentions and uncover distant yet relevant nodes. RAPTOR (Sarthi et al., 2024) introduces a tree-based hierarchical organization that recursively summarizes evidence at different levels, balancing efficiency with global context coverage. GFM-RAG (Luo et al., 2025b) leverages graph neural networks (GNNs) to encode structural dependencies and propagate information across neighbors, thereby capturing higher-order relations that are crucial for multi-hop reasoning. Meanwhile, GraphRAG (Edge et al., 2024) applies community detection to partition large knowledge graphs into semantically coherent clusters, reducing noise and emphasizing connections among related entities.

**Iterative Retrieval.** Iterative retrieval has been widely explored as a means to enhance RAG by dynamically refining the evidence set in multiple rounds of reasoning and retrieval. IRCOT (Trivedi et al., 2022a) interleaves retrieval with chain-of-thought reasoning, allowing the model to iteratively issue new queries guided by intermediate reasoning steps. IRGS (Shao et al., 2023) emphasizes the synergy between retrieval and generation, where iterative refinement of both components leads to stronger evidence grounding. TOG (Sun et al., 2023) extends iterative retrieval to the graph setting, enabling LLMs to progressively navigate and reason over knowledge graphs. GCOT (Jin et al., 2024) integrates graph structures into the chain-of-thought process, combining step-by-step reasoning with structured retrieval to capture multi-hop dependencies more effectively. Collectively, these approaches demonstrate the importance of iterative retrieval as a general strategy to mitigate missing evidence and strengthen multi-step reasoning in complex QA tasks. Recent systems such as GFM-RAG (Luo et al., 2025b) and HippoRAG (Jimenez Gutierrez et al., 2024) incorporate only limited use of iterative retrieval. Nevertheless, how iterative retrieval strategies function within the GraphRAG framework remains largely unexplored.

| Case Study | |
|---|---|
| Question: | *At what intersection was the former home of the wooden roller coaster now located at Six Flags Great America in Gurnee, Illinois located?* |
| Gold Answer: | North Avenue and First Avenue |
| **Method** | **Retrieved Evidence → Prediction** |
| HippoRAG2 w/ IRCOT | (Retrieved) *Little Dipper*: "relocated from Kiddieland Amusement Park." (Retrieved) *Kiddieland Amusement Park*: "located at North Avenue and First Avenue." → **Correct**: North Avenue and First Avenue |
| HippoRAG2 | (Retrieved) *Little Dipper*: "relocated from Kiddieland Amusement Park." (Missed) No document mentioning Kiddieland's intersection (bridge fact missing). → **Incorrect**: Cannot infer the intersection |

Figure 7: Case study showing the importance of retrieving *bridge facts*. With IRCOT (top), the retriever surfaces Kiddieland's location, enabling the correct answer. Without it (bottom), the reasoning chain breaks.

| Methods | Recall@5 | Recall@10 |
|---|---|---|
| RAPTOR w/ IRCOT | 0.7584 | 0.8134 |
| RAPTOR w/ Our | 0.8110 | 0.8624 |

Table 3: Retrieval performance on Musique

## A.3 ALGORITHM

The overall process of **Bridge-Guided Dual-Thought-based Retrieval (BDTR)** is summarized in Algorithm 1. It integrates the two key components introduced above: Dual-Thought-based Retrieval (DTR) for coverage expansion and Bridge-Guided Evidence Calibration (BGEC) for ranking calibration. In particular, Lines 3–6 correspond to DTR, where dual thoughts generate complementary retrieval signals and their results are merged into a shared pool. Lines 7–11 correspond to BGEC, where bridge-supporting documents are identified via the reasoning chain and re-ranked, followed by statistical filtering to form the final evidence set.

---

**Algorithm 1:** BDTR: Bridge-Guided Dual-Thought-based Retrieval

---

**Input:** question $Q$, backbone retriever $f_{\text{ret}}$, number of iterations $R$
**Output:** final document set $\mathcal{D}_{\text{final}}$

**1:** $P_0 \leftarrow f_{\text{ret}}(Q)$ ;                    // Initial retrieval
**for** $t = 1$ **to** $R$ **do**
   **2:** Generate two complementary queries $q_t^{\text{FT}}, q_t^{\text{ST}}$ from current pool;
   **3:** $D_t^{\text{FT}} \leftarrow f_{\text{ret}}(q_t^{\text{FT}})$;
   **4:** $D_t^{\text{ST}} \leftarrow f_{\text{ret}}(q_t^{\text{ST}})$;
   **5:** $P_t \leftarrow P_{t-1} \cup D_t^{\text{FT}} \cup D_t^{\text{ST}}$;
   **6:** Update scores and resort: $s_t(d) \leftarrow \max(s_{t-1}(d), \hat{s}(d \mid q_t^{\text{FT}}), \hat{s}(d \mid q_t^{\text{ST}}))$ for $d \in P_t$;

**7:** Generate reasoning chain $RC$ from final pool $P_R$;
**8:** $\mathcal{G} \leftarrow \{d \in P_R \mid \text{Verifier}(d, RC) = 1\}$ ;       // Bridge docs selected by LLM
**9:** Promote $\mathcal{G}$ to the top of $P_R$;
**10:** Compute $\mu, \sigma$ from scores of top-50 docs in $P_R$;
**11:** $\mathcal{D}_{\text{final}} \leftarrow \{d \in P_R \mid s(d) \geq \mu + \sigma\}$, ensuring $|\mathcal{D}_{\text{final}}| \geq 5$;
**return** $\mathcal{D}_{final}$;

---

## A.4 EFFICIENCY ANALYSIS

Table 4 reports the retrieval runtime on Musique in comparison with HippoRAG2.

Compared with IRCOT, our method introduces dual-thought generation and bridge-based evidence calibration, which increases the computational burden and leads to a modest rise in latency (1.7h vs. 1.1h). To address this overhead, we optimize the framework by parallelizing the retrieval pipeline across different queries. Concretely, each query is still processed sequentially to preserve reasoning consistency, but multiple queries are dispatched concurrently via a thread pool. This design reduces the latency of GPT calls and graph-based retrieval I/O, which are the dominant bottlenecks. As a result, the total running time is reduced from 1.7h to 0.3h, yielding a 5.7× speedup.

|                        | Musique |
| ---------------------- | ------- |
| IRCOT                  | 1.1h    |
| Our                    | 1.7h    |
| Our w/ parallelization | 0.3h    |

Table 4: Comparison of running time on Musique with HippoRAG2.

### A.5 ABLATION STUDY

To understand the contribution of each component, we conduct an ablation study on the two core modules of BDTR: Dual-Thought-based Retrieval (DTR) and Bridge-Guided Evidence Calibration (BGEC). The results are summarized in Table 5.

We observe that DTR alone brings a clear performance boost, improving EM by 23.6% and F1 by 19.4% on dataset Musique. This demonstrates the effectiveness of generating complementary thoughts to enlarge the evidence frontier. BGEC further enhances performance by recalibrating the ranking based on the reasoning chain, improving EM by 31.1% and F1 by 25.8%. When both modules are combined, BDTR achieves the best results, with overall gains of 34.8% in EM and 29.2% in F1. These findings validate that both DTR and BGEC are essential and complementary, jointly contributing to the strong performance of BDTR.

| Methods         | EM    | F1    |
| --------------- | ----- | ----- |
| RAPTOR          | 0.296 | 0.418 |
| RAPTOR w/ DTR   | 0.366 | 0.499 |
| RAPTOR w/ BGEC  | 0.388 | 0.526 |
| RAPTOR w/ BDTR  | 0.399 | 0.540 |

Table 5: Ablation Study.

### A.6 PROMPT EXAMPLE

In this section, we provide the prompt examples for iterative methods: our method, IRCOT, ToG, GCOT, IRGS. They are illustrated in Fig 8, Fig 9, Fig 10, Fig 11, Fig 12 and Fig 13.

---

**Prompts**

You are an intelligent assistant skilled in multi-hop reasoning across multiple documents. For every turn, produce two outputs:
**Fast Thought**: A direct follow-up question asking for the missing fact in plain form.
**Slow Thought**: A follow-up question phrased with a reasoning flavor, showing part of the solution path inside the question itself.
**Rules:**

- Fast Thought must be short and direct (e.g., "Where was X born?").

- Slow Thought must explicitly include an additional bridging entity or relation, not just a rephrasing of the Fast Thought.

- Slow Thought should demonstrate a reasoning chain style, embedding at least one bridge or context element that connects to the target fact.

- Output only the two thoughts, and no other explanations.

- Goal: provide both a direct query and a reasoning-flavored query that retrieve complementary bridge documents.

---

Figure 8: An Example Prompt of Dual-Thought Generation.

**Prompts**

You are a careful reasoning verifier.
**Reasoning Chain (arrow format, ≥4 nodes):** {reasoning chain}
**Candidate Documents (Top-30 as currently ranked):** {listing}
**Your task:**

- Identify which documents provide direct evidence for one or more **critical nodes or bridges** in the reasoning chain (e.g., the key missing fact, entity–relation links, or decisive constraints).

- **Do not** bias toward the current rank. If a lower-ranked document (e.g., Doc 30) strongly supports a critical node or a bridging relation, you **must** include it.

- Favor documents that:
    - explicitly answer the Direct Question node(s),
    - provide the bridge relation in the Graph Thought,
    - connect multiple nodes in the chain (multi-node coverage),
    - resolve ambiguities or disambiguate entities.

- Output **strict JSON only** with a single key `"covered_doc_indices"` as a list of 1-based integers.

- Order indices by **support/impact strength** (strongest first), not by rank.

**Example (STRICT JSON):** {{"covered_doc_indices":[17,3,28]}}
Return STRICT JSON only. No explanations.

Figure 9: An Example Prompt of Bridge-based Evidence Calibration.

**Prompts**

You serve as an intelligent assistant, adept at facilitating users through complex, multi-hop reasoning across multiple documents. This task is illustrated through demonstrations, each consisting of a document set paired with a relevant question and its multi-hop reasoning thoughts.
**Your task:** Generate **one reasoning thought for the current step**.

- Do not generate the entire reasoning chain at once.

- At each step, provide only a single intermediate thought that advances the reasoning.

- If you believe you have reached the final step, begin your response with: `"So the answer is:"`.

Figure 10: An Example Prompt of IRCOT.

**Prompts**

You serve as an intelligent assistant, adept at facilitating users through complex, multi-hop reasoning across multiple documents. This task is illustrated through demonstrations, each consisting of a document set paired with a relevant question and its multi-hop reasoning thoughts.
**Your task:** Evaluate whether the given information is sufficient to answer the question.

- If the evaluation is positive, start the response with: `"So the answer is:"`.

- Otherwise, explain what additional information would be required.

Figure 11: An Example Prompt ToG

**Prompts**

You serve as an intelligent assistant, adept at facilitating users through complex, multi-hop reasoning across multiple documents. This task is illustrated through demonstrations, each consisting of a document set paired with a relevant question and its multi-hop reasoning thoughts.

**Your task:** Think step by step about what additional information is required for the current step.

- Do not generate the full reasoning or the final answer at once.
- At each step, only articulate what is still missing or what bridge evidence should be retrieved next.
- If you reach what you believe to be the final step, begin your response with: `"So the answer is:"`.

Figure 12: An Example Prompt of GCOT.

**Prompts**

Based on the following documents, answer the question with concise reasoning and a final answer. Keep the reasoning under 100 English words.
**Documents:** {context}
**Question:** {query}
**Please provide:**

1. Brief reasoning based on the documents
2. Final answer

Figure 13: An Example Prompt of IRGS

