# OpenReview forum: "Beyond Static Retrieval: Opportunities and Pitfalls of Iterative Retrieval in GraphRAG"
_ICLR.cc/2026/Conference — ICLR 2026 Conference Withdrawn Submission_

### Official Review · Reviewer_MBGR · 2025-10-22

**Soundness:** 3
**Presentation:** 2
**Contribution:** 2
**Rating:** 2
**Confidence:** 4

**Summary:**

The paper reveals that iterative retrieval can significantly enhance multi-hop reasoning in GraphRAG by uncovering bridging evidence, and introduces BDTR, a dual-thought and bridge-guided framework that makes this process more accurate and efficient.

**Strengths:**

1. This paper conducts a comprehensive preliminary study that systematically examines the opportunities and challenges of iterative retrieval in GraphRAG, providing strong empirical motivation and clear research directions for the subsequent method design and experiments.

2. The proposed BDTR framework effectively enhances GraphRAG by combining dual-thought retrieval and bridge-guided evidence calibration, which improves both retrieval and reasoning in multi-hop QA tasks.

**Weaknesses:**

1.	**Narrow motivation scope.** In Section 2 (Preliminary Study), the identified opportunities and pitfalls are derived from empirical studies on multi-hop QA datasets. However, since these datasets are deliberately constructed and highly structured, the motivation and findings are confined to a narrow task setting. This restricts the generalizability of BDTR to broader retrieval-augmented tasks such as multi-turn dialogue, contextual summarization, fact checking and other open-ended querying, where multiple pieces of evidence from a corpus must also be aggregated to support a response. These tasks are not predefined with fine-grained question types like "bridging" or "comparison" but worth handling.

2.	**Lack of case studies.** The paper does not provide case studies or qualitative examples to illustrate how iterative retrieval operates in practice, or to demonstrate why BDTR outperforms other iterative retrieval methods in a more interpretable manner. This omission weakens readers’ intuition about the mechanism behind BDTR’s improvements.

3.	**Sparse integration with GraphRAG.** In Section 3.1 (DTR), the backbone retriever merely returns documents from the candidate pool, with GraphRAG’s sole contribution being the provision of retrieval scores. Consequently, BDTR lacks explicit interaction with entities or relations within the underlying graph. Furthermore, the baseline implementation of ToG, which should involve graph-based search and pruning, is simplified into a document-level retrieval process. This oversimplification of graph integration in the experimental setup arguably dilutes the contribution of BDTR within the GraphRAG framework.

4.	**Omission of embedding-based dense retrievers.** The paper does not explore BDTR’s performance when paired with embedding-based dense retrievers. It remains unclear what concrete advantages a GraphRAG-based retriever provides over a dense retriever, given that both can supply the retrieval scores required by BDTR, while the latter is typically more efficient and computationally cheaper.

**Questions:**

As discussed in Weakness.

---

> ### Author Response · Authors · 2025-11-24
> **Responses to Reviewer MBGR**
>
> We appreciate the reviewer’s valuable feedback and provide our responses to each comment below.
>
> **W1:** Narrow motivation scope. In Section 2 (Preliminary Study), the identified opportunities and pitfalls are derived from empirical studies on multi-hop QA datasets. However, since these datasets are deliberately constructed and highly structured, the motivation and findings are confined to a narrow task setting. This restricts the generalizability of BDTR to broader retrieval-augmented tasks such as multi-turn dialogue, contextual summarization, fact checking and other open-ended querying, where multiple pieces of evidence from a corpus must also be aggregated to support a response. These tasks are not predefined with fine-grained question types like "bridging" or "comparison" but worth handling.
>
> A1: We conduct additional experiments on the Medical datasets from GraphRAG-Bench [1]. These datasets cover several task types, including complex reasoning, contextual summarization, and fact retrieval. We use both LLM-as-a-judge and ROUGE-L to measure performance, following the same evaluation metric as the original paper. As shown in the table, our method consistently outperforms IRCOT across all task categories.
>
> | Task                 | IRCOT LLM_ACC | Ours LLM_ACC | IRCOT ROUGE-L | Ours ROUGE-L |
> |----------------------|----------------|---------------|----------------|---------------|
> | Complex Reasoning    | 0.8389         | **0.8566**        | 0.3577         | **0.3600**        |
> | Contextual Summarize | 0.8166         | **0.8408**        | 0.3569         | **0.3715**        |
> | Fact Retrieval       | 0.8042         | **0.8342**       | 0.3865         | **0.3933**        |
>
> [1] Xiang Z, Wu C, Zhang Q, et al. When to use graphs in rag: A comprehensive analysis for graph retrieval-augmented generation[J]. arXiv preprint arXiv:2506.05690, 2025.
>
> **W2:** Lack of case studies. The paper does not provide **case studies** or qualitative examples to illustrate how iterative retrieval operates in practice, or to demonstrate why BDTR outperforms other iterative retrieval methods in a more interpretable manner. This omission weakens readers’ intuition about the mechanism behind BDTR’s improvements.
>
> A2: We provide both successful and failure cases to illustrate how our method behaves in practice.
> # Successful Case
>
> | Question | What year did the Governor of the city where the basilica named after the same saint as the one that Mantua Cathedral is dedicated to die? |
> |---------|------------------------------------------------------------------------------------------------------------------------------------------------------------------------------------------------------------------------------------------------------------------|
> | Gold Answer | 1952 |
> | Reasoning Chain | (Mantua Cathedral) --[dedicated to]--> (Saint Peter) --[associated city]--> (Vatican City) --[governed by]--> (Governor of Vatican City) --[died in]--> (time) |
> | Prediction Answer | 1952. |
> | Find the bridges | **Bridge 1:** Confirmed that Mantua Cathedral is dedicated to Saint Peter. **Bridge 2:** Identified that St. Peter's Basilica is located in Vatican City. **Bridge 3:** Determined that the Governor of Vatican City (Camillo Serafini) held the post until his death in 1952. |
>
>
>
> # Failure Case
> | Question | An Indy car race was held in the capital of the state where the performer of Mingus Plays Piano was born. Who won the race? |
> |---------|------------------------------------------------------------------------------------------------------------------------------------------------------------------------------------------------------------------------------------------------------------------|
> | Gold Answer | Mario Andretti |
> | RC (Reasoning Chain) | (Charles Mingus) --[born in]--> (Nogales, Arizona) --[race held in]--> (Indianapolis, Indiana) --[winner of]--> (Indy car race in Indianapolis) |
> | Prediction Answer | The winner of the race is not provided in the text. |
> | Missing bridge | The model correctly identified that Charles Mingus was born in Arizona but failed to map the state capital correctly. The capital of Arizona is Phoenix, not Indianapolis, causing the model to search for races in Indianapolis instead of the Phoenix Grand Prix, where Mario Andretti won his final race.

---

> ### Author Response · Authors · 2025-11-24
> **Responses to Reviewer MBGR**
>
> **W3:** Sparse integration with GraphRAG. In Section 3.1 (DTR), the backbone retriever merely returns documents from the candidate pool, with GraphRAG’s sole contribution being the provision of retrieval scores. Consequently, BDTR lacks explicit interaction with entities or relations within the underlying graph. Furthermore, the baseline implementation of ToG, which should involve graph-based search and pruning, is simplified into a document-level retrieval process. This oversimplification of graph integration in the experimental setup arguably dilutes the contribution of BDTR within the GraphRAG framework.
>
> A3: We would like to clarify that our method is fundamentally grounded in the principles of GraphRAG. Although our iterative retrieval module does not operate on an explicitly graph, the reasoning stage is graph-guided. The Fast Thought (FT), Slow Thought (ST), and Reasoning Chain (RC) are not free-form textual generations. Instead, the model is prompted to treat the query and retrieved passages as an implicit local graph, and its outputs are constrained by this latent structure. In this sense, FT, ST, and RC function as linguistic interfaces for topological exploration, edge validation, and path construction.
>
> First, the RC requires the model to produce an explicit entity–relation chain, which forces it to extract entities and relations from the retrieved context and commit to a structured multi-hop path. This imposes graph-style supervision rather than allowing unconstrained paraphrasing.
>
> Second, the FT is not a simple reformulation of the original question. It is a targeted query for the next missing node along the chain established in the RC, aligning the model with stepwise graph traversal and path extension rather than purely textual reasoning.
>
> Third, the ST requires the model to justify why a specific bridge entity is needed, making it explicitly articulate how this intermediate node advances the relational path. This justification is inherently topological and operationalizes graph-based neighborhood expansion, even without an explicitly instantiated graph.
>
> Here is our prompt.
> # **prompt**
>
> ---
>
> You are a "Graph Reasoning Agent". Your goal is to navigate a hidden Knowledge Graph to answer complex questions.
> You must identify the "Bridge Entities" that connect the User's Question to the Answer.
>
>
> For every turn, analyze the provided text and produce three specific outputs:
>
> 1. **Reasoning Chain (The Map):**
>    - Construct a valid entity path based on the context.
>    - Format: (Entity A) --[relation]--> (Entity B) --[relation]--> (Target).
>    - CONSTRAINT: Every entity in the chain must be explicitly related to the context. Do not hallucinate links.
>
> 2. **Fast Thought (The Next Step):**
>    - A specific, short question targeting the *next missing node* in your chain.
>    - Example: "Where is [Entity B] located?" (instead of "Where is the answer?")
>
> 3. **Slow Thought (The Topology):**
>    - Explain *why* you are asking the Fast Thought based on graph structure.
>    - Format:
>      **"I have identified [Entity A]. To find [Target], I must first retrieve the neighbor [Entity B] which acts as a bridge."**
>
>
>
> **Rules:**
> - Output format: exactly three lines in this order:
>   1) Reasoning Chain: ...
>   2) Fast Thought: ...
>   3) Slow Thought: ...
>
> ---
>
> **W4:** Omission of embedding-based dense retrievers.
>
> A4: Thanks for your suggestion. I conduct experiments using embeeding-based dense retriever, such as NV-Embed-v2. The results in table demonstrate that our method also can improve the performance for dense retriever. In HotpotQA, the dense retriever (NV-Embed-v2+BDTR) slightly outperforms GFM-RAG+BDTR, suggesting that a dense embedding–based retriever is already sufficient for datasets with relatively linear reasoning chains. However, on 2WikiMultihopQA, GFM-RAG+BDTR achieves a substantially higher score than the dense retriever. This indicates that graph-based retrieval remains highly beneficial for datasets requiring more complex multi-hop reasoning.
>
> | Method                 | HotpotQA EM | HotpotQA F1 | 2wikimultihopqa EM | 2wikimultihopqa F1 |
> |------------------------|-------------|-------------|---------------------|---------------------|
> | NV-Embed-v2            | 0.590           | 0.7348           | 0.528               | 0.597               |
> | NV-Embed-v2+BDTR | **0.602**      | **0.7507**      | 0.668               | 0.764              |
> | GFM-RAG+BDTR     | 0.585       | 0.7462      | **0.726**               | **0.805**             |

---

### Official Review · Reviewer_D2Gw · 2025-10-30

**Soundness:** 2
**Presentation:** 2
**Contribution:** 2
**Rating:** 2
**Confidence:** 5

**Summary:**

This paper investigates iterative retrieval strategies within the GraphRAG framework for multi-hop question answering. The authors conduct preliminary studies to identify opportunities and pitfalls of iterative retrieval, and compare different strategies.

**Strengths:**

**Clear motivation and empirical investigation:** The paper systematically analyzes how iterative retrieval affects performance on different question types (Bridge vs. Comparison), highlighting both benefits and limitations.

**Comparative analysis of strategies:** Evaluating multiple iterative retrieval strategies and suggesting possible combination or adaptive selection could be valuable.

**Weaknesses:**

Shown in **Questions**.

**Questions:**

**1. Lack of in-depth analysis**
FT and ST are both generated by the model through prompts. This process is highly dependent on the model's capabilities and has low controllability. The experiments in the paper do not clearly demonstrate the specific reasoning behaviors under the BDTR strategy. I suspect that such a simple expansion strategy could introduce unexpected noise information, which may reduce retrieval and reasoning precision. **Quantitative results on retrieval accuracy are needed to substantiate the claimed improvements in retrieval. In addition, could the authors provide several examples of both successful and failure cases to illustrate how BDTR behaves in practice?**

**2. The conclusions in the "Preliminary Studies" section are not surprising**
The conclusions presented here are widely recognized phenomena in the field, and they do not provide novel insights. It would be helpful if the authors could provide more original analysis or novel observations based on your work.

**3. Missing important details**
> Retrieval scores are generated by GraphRAG backbones.

The pipeline heavily relies on these scores, but the paper does not explain how these scores are computed. The authors should provide more details on the score generation process and their implementations.

**4. Static iterations**
Although the authors introduce the "fast" and "slow" thought process in multi-round RAG to address the problem of overthinking when switching between complex and simple tasks,
The method in this paper fixes the number of iterations in advance. For simple problems that require just one step, additional iterations still introduce unnecessary noise. This contradicts the goal of "Beyond Static Retrieval," as it does not adapt dynamically to different cases.

**5. Limited innovation**
Apart from the points mentioned above, the innovation of the approach is rather limited. The overall process still revolves around prompt-based multi-round query rewriting followed by LLM self-validation.

---

> ### Author Response · Authors · 2025-11-24
> **Responses to Reviewer D2Gw**
>
> We appreciate the reviewer’s valuable feedback and provide our responses to each comment below.
>
> **W1:**  I suspect that such a simple expansion strategy could introduce unexpected noise information, which may reduce retrieval and reasoning precision. Quantitative results on retrieval accuracy are needed to substantiate the claimed improvements in retrieval. In addition, could the authors provide several examples of both successful and failure cases to illustrate how BDTR behaves in practice?
>
> A1: We appreciate the reviewer’s concern about potential noise in Dual-Thought-based Retrieval (DTR). DTR is designed as a recall-first, precision-controlled process: Fast Thought (FT) and Slow Thought (ST) expand the search space to increase the chance of retrieving bridge candidates. This expansion is not arbitrary—FT and ST are graph-guided, functioning as linguistically expressed queries derived from an implicit local graph inferred from the question and retrieved passages.
>
> Our method is fundamentally grounded in GraphRAG principles. FT, ST, and the Reasoning Chain (RC) guide the model to interpret the context as a latent entity–relation graph. These components serve as interfaces for topological exploration, edge checking, and path construction, ensuring that retrieval expansion remains structurally informed rather than noise-driven.
>
> To further preserve precision after expansion, we apply Bridge-Guided Evidence Consolidation (BGEC), which reranks and filters the retrieved candidates by promoting passages that contain plausible bridge evidence and suppressing irrelevant noise. This two-stage design (DTR → BGEC) ensures that expansion improves recall without degrading end-to-end reasoning accuracy.
>
> We then provide quantitative retrieval results, which show that our method achieves higher retrieval accuracy.
>
> ### 2WikiMultiHopQA
>
> | Method | Recall@1 | Recall@2 | Recall@5 |
> |--------|----------|----------|----------|
> | **IRCOT** | 0.4158 | 0.7935 | 0.9860 |
> | **Ours** | 0.4323 | 0.8570 | 0.9932 |
>
> ---
>
> ### MuSiQue
>
> | Method | Recall@1 | Recall@2 | Recall@5 |
> |--------|----------|----------|----------|
> | **IRCOT** | 0.3172 | 0.5571 | 0.7891 |
> | **Ours** | 0.3260 | 0.5671 | 0.8009 |
>
> Finally, we provide both successful and failure cases to illustrate how our method behaves in practice.
> # Successful Case
>
> | Question | What year did the Governor of the city where the basilica named after the same saint as the one that Mantua Cathedral is dedicated to die? |
> |---------|------------------------------------------------------------------------------------------------------------------------------------------------------------------------------------------------------------------------------------------------------------------|
> | Gold Answer | 1952 |
> | Reasoning Chain | Mantua Cathedral is dedicated to Saint Peter -> Governor of Vatican City is related to Saint Peter -> What year did the Governor of Vatican City die? |
> | Prediction Answer | 1952. |
> | Find the bridges | **Bridge 1:** Confirmed that Mantua Cathedral is dedicated to Saint Peter. **Bridge 2:** Identified that St. Peter's Basilica is located in Vatican City. **Bridge 3:** Determined that the Governor of Vatican City (Camillo Serafini) held the post until his death in 1952. |
>
>
>
> # Failure Case
> | Question | An Indy car race was held in the capital of the state where the performer of Mingus Plays Piano was born. Who won the race? |
> |---------|------------------------------------------------------------------------------------------------------------------------------------------------------------------------------------------------------------------------------------------------------------------|
> | Gold Answer | Mario Andretti |
> | RC (Reasoning Chain) | Charles Mingus was born in Nogales, Arizona -> The race was held in Indianapolis, Indiana -> Who won the Indy car race in Indianapolis? |
> | Prediction Answer | The winner of the race is not provided in the text. |
> | Missing bridge | The model correctly identified Charles Mingus was born in Arizona, but failed to correctly map the capital. The capital of Arizona is Phoenix, not Indianapolis. The model hallucinated that Mingus was born in Indiana or conflated "Indy car" with "Indiana," leading it to search for Indianapolis races instead of the Phoenix Grand Prix, where Mario Andretti won his final race. |

---

> ### Author Response · Authors · 2025-11-24
> **Responses to Reviewer D2Gw**
>
> **W2:** The conclusions in the "Preliminary Studies" section are not surprising The conclusions presented here are widely recognized phenomena in the field, and they do not provide novel insights. It would be helpful if the authors could provide more original analysis or novel observations based on your work.
>
> A2: To the best of our knowledge, no prior work has systematically examined how iterative retrieval behaves within the GraphRAG framework. Our preliminary experiments fill this gap and show that iterative retrieval in GraphRAG leads to significant performance gains. At the same time, we find that the identification of bridge evidence itself remains a critical bottleneck for multi-hop reasoning. Based on these observations, we propose BDTR, which contributes both (i) a new understanding of iterative retrieval in GraphRAG, and (ii) a concrete two-module framework that operationalizes this insight.
>
> To further understand the iterative retrieval, we will conduct a deeper analysis of the distinctions between GraphRAG and standard RAG. We run experiments using embedding-based dense retrievers such as NV-Embed-v2. The results in the table show that our method also improves performance in this setting. On HotpotQA, the dense retriever (NV-Embed-v2 + BDTR) slightly outperforms GFM-RAG + BDTR, suggesting that dense embedding–based retrieval is sufficient for datasets with relatively linear reasoning chains. However, on 2WikiMultihopQA, GFM-RAG + BDTR achieves a substantially higher score than the dense retriever, indicating that graph-based retrieval remains highly advantageous for datasets requiring more complex multi-hop reasoning.
>
> | Method                 | HotpotQA EM | HotpotQA F1 | 2wikimultihopqa EM | 2wikimultihopqa F1 |
> |------------------------|-------------|-------------|---------------------|---------------------|
> | NV-Embed-v2            | 0.590           | 0.7348           | 0.528               | 0.597               |
> | NV-Embed-v2+BDTR | 0.602       | 0.7507      | 0.668               | 0.764              |
> | GFM-RAG+BDTR     | 0.585       | 0.7462      | 0.726               | 0.805             |
>
> **W3:** . Missing important details: retrieval scores are generated by GraphRAG backbones.
> A3: Retrieval scores are generated by GraphRAG backbones. I will provide the details, taking HippoRAG2 as an example. In this framework, the document score arises from a single Personalized PageRank computation: the query assigns weights to entity nodes that appear in the most relevant facts, while passages that are semantically similar to the query receive Dense Passage Retrieval (DPR)-based weights. These weights jointly define the reset distribution for PageRank, which then propagates over the entity–passage graph until convergence.
>
> Formally, the weight assigned to an entity $e$ is
> $$
> w^{(e)} = \frac{s_f}{c_e},
> $$
> where $s_f$ is the fact–query similarity and $c_e$ is the number of corpus chunks containing $e$. Let $w^{(p)}$ denote the DPR-based weight for passage $p$. The reset vector is
> $$
> r_i = w_i^{(e)} + w_i^{(p)}.
> $$
>
> Personalized PageRank then produces passage relevance scores as
> $$
> \mathrm{score}(p) = \mathrm{PPR}_p,
> $$
> where $\mathrm{PPR}_p$ is the PageRank value of passage $p$ under reset distribution $r$.
>
> **W4:** Although the authors introduce the "fast" and "slow" thought process in multi-round RAG to address the problem of overthinking when switching between complex and simple tasks, The method in this paper fixes the number of iterations in advance. For simple problems that require just one step, additional iterations still introduce unnecessary noise. This contradicts the goal of "Beyond Static Retrieval," as it does not adapt dynamically to different cases.
>
> A4: We emphasize that the choice of two retrieval iterations is not arbitrary—it is supported by our empirical analysis (Section 2.5), which shows that two iterations provide a good balance between retrieval gains and computational cost. To ensure fair comparisons across different datasets and baselines, we fixed the number of iterations to two in our main experiments. Furthermore, the BDTR framework is fully compatible with adaptive stopping strategies.

---

> ### Author Response · Authors · 2025-11-24
> **Responses to Reviewer D2Gw**
>
> **W5:** Limited innovation Apart from the points mentioned above, the innovation of the approach is rather limited. The overall process still revolves around prompt-based multi-round query rewriting followed by LLM self-validation.
>
>
> A5: To the best of our knowledge, prior research has not systematically examined how iterative retrieval operates within the GraphRAG framework. Our empirical analysis reveals that the primary bottleneck in GraphRAG stems from the ranking of bridge evidence. Building on this insight, BDTR provides both a new conceptual understanding of iterative retrieval in GraphRAG and a concrete two-module framework that operationalizes this understanding—moving beyond standard prompt-based query rewriting.
>
> Specifically, BDTR is explicitly graph-guided. The Fast Thought (FT), Slow Thought (ST), and Reasoning Chain (RC) are not generated by the LLM in an unconstrained, free-form textual manner. Instead, the model is prompted to interpret the query and retrieved passages as an implicit local graph, and its generations are guided by this latent graph structure. This design forces the model to adopt structured, graph-like reasoning. In this sense, FT, ST, and RC serve as linguistic interfaces for topological exploration, edge validation, and path construction.
>
> Here is our prompt for generating FT, ST and RC.
> # **prompt**
>
> ---
>
> You are a "Graph Reasoning Agent". Your goal is to navigate a hidden Knowledge Graph to answer complex questions.
> You must identify the "Bridge Entities" that connect the User's Question to the Answer.
>
>
> For every turn, analyze the provided text and produce three specific outputs:
>
> 1. **Reasoning Chain (The Map):**
>    - Construct a valid entity path based on the context.
>    - Format: (Entity A) --[relation]--> (Entity B) --[relation]--> (Target).
>    - CONSTRAINT: Every entity in the chain must be explicitly related to the context. Do not hallucinate links.
>
> 2. **Fast Thought (The Next Step):**
>    - A specific, short question targeting the *next missing node* in your chain.
>    - Example: "Where is [Entity B] located?" (instead of "Where is the answer?")
>
> 3. **Slow Thought (The Topology):**
>    - Explain *why* you are asking the Fast Thought based on graph structure.
>    - Format:
>      **"I have identified [Entity A]. To find [Target], I must first retrieve the neighbor [Entity B] which acts as a bridge."**
>
>
>
> **Rules:**
> - Output format: exactly three lines in this order:
>   1) Reasoning Chain: ...
>   2) Fast Thought: ...
>   3) Slow Thought: ...
>
> ---
>
> Moreover, through a parallelized implementation, we further improve the efficiency of iterative retrieval by 5.7×, demonstrating that BDTR achieves both conceptual novelty and practical scalability.

---

> > ### Comment · Reviewer_D2Gw · 2025-11-27
> >
> > Thank you for the response. I have decided to keep my score as it is.

---

### Official Review · Reviewer_5CKK · 2025-10-30

**Soundness:** 4
**Presentation:** 3
**Contribution:** 4
**Rating:** 8
**Confidence:** 4

**Summary:**

This paper systematically analyzes the iterative retrieval mechanism within GraphRAG. The authors identify a key bottleneck in GraphRAG: while graph structures support multi-hop reasoning, static retrieval often fails to surface critical *bridge documents*, the intermediate evidence that connects disjoint entities.

Through thorough experiments across multiple dimensions, such as effectiveness, question types,  recall, number of rounds, and complementarity, this paper demonstrates that iterative retrieval can improve performance in complex multi-hop questions, help promote bridge documents into leading ranks, and different strategies offer complementary strengths. The experiments also show the pitfalls that naive expansion often introduces noise, iterations offer little benefit on simple comparison questions, and many evidences still fail to appear within the leading positions.

Based on these observations, this paper proposes Bridge-Guided Dual-Thought-based Retrieval (BDTR), which combines dual-thought generation with bridge-aware reranking to better promote bridge evidence into leading positions. BDTR shows consistent improvements across multiple GraphRAG backbones and datasets, offering both empirical insights and a practical solution for enhancing GraphRAG systems.

**Strengths:**

- This paper systematically studies the iterative retrieval mechanism in GraphRAG, offering a empirical insights for enhancing GraphRAG systems.
- The proposed BDTR framework is novel, solidly grounded in the experiments and observations on iterative retrieval mechanisms, and is clearly described in the paper.
- Both in the investigation of mechanisms and the validation of BDTR, the experiments are thoroughly conducted and robust, with conclusions sufficiently supported by evidences.

**Weaknesses:**

BDTR exhibits a strong reliance on the LLM, so the LLM's performance may impact the final results and computation costs. However, the paper lacks relevant experiments across different LLMs.

**Questions:**

Refer to Weaknesses, will the performance of LLMs used in BDTR significantly affect the experimental results?

---

> ### Author Response · Authors · 2025-11-24
> **Responses to Reviewer 5CKK**
>
> We appreciate the reviewer’s valuable suggestions and provide our responses below.
>
> **W1:** BDTR exhibits a strong reliance on the LLM, so the LLM's performance may impact the final results and computation costs. However, the paper lacks relevant experiments across different LLMs.
>
> A1: Thank you for the helpful suggestion. We conducted additional comparisons using the DeepSeek model. The results show that our method consistently outperforms the baseline across different LLMs.
> | Method   | HotpotQA EM | HotpotQA F1 | Musique EM | Musique F1 |
> |----------|-------------|-------------|---------------------|---------------------|
> | IRCOT | 0.649       | 0.7794      | 0.419               | 0.5451              |
> | Ours    | **0.658**       | **0.7882**     | **0.469**           | **0.5831**          |

---

### Official Review · Reviewer_jjM6 · 2025-10-31

**Soundness:** 3
**Presentation:** 3
**Contribution:** 2
**Rating:** 4
**Confidence:** 4

**Summary:**

This paper presents Bridge-Guided Dual-Thought-based Retrieval (BDTR), a novel iterative framework for GraphRAG that systematically promotes bridge evidence into leading retrieval ranks, thereby overcoming the limitations of static retrieval and significantly improving multi-hop reasoning performance.

**Strengths:**

This paper effectively addresses the long-standing challenge of missing or under-ranked bridge evidence in GraphRAG, which often breaks multi-hop reasoning chains. Its observations are strongly supported by extensive and systematic experiments across multiple datasets and GraphRAG backbones, demonstrating consistent and significant performance gains.

**Weaknesses:**

1. The experimental coverage and task diversity are limited. Since all the test datasets are derived from Wikipedia, the paper provides insufficient evidence of generalization and robustness to open-domain, multi-source, or non-encyclopedic corpora such as enterprise knowledge bases.

2. The paper’s practical value for broader complex reasoning tasks remains uncertain. While BDTR effectively targets the “bridge evidence” problem in multi-hop QA, its design is highly specialized for this scenario and may not generalize well to other reasoning settings—such as temporal, causal, or commonsense reasoning—where evidence relationships are more abstract and cannot be explicitly modeled through bridge documents.

3. The proposed approach underutilizes the structured nature of GraphRAG. Although GraphRAG is designed to leverage entity–relation graphs for structured reasoning, BDTR primarily operates at the text retrieval and reranking level, without explicitly incorporating or reasoning over the underlying graph topology, edge semantics, or relational paths, missing opportunities to exploit the full potential of graph-based representations.

4. The paper lacks a detailed analysis of failure cases where the Dual-Thought mechanism fails to retrieve the bridge documents. Without examining such negative examples, it remains unclear how robust the overall retrieval system is under challenging scenarios—e.g., when bridge evidence is implicitly expressed, sparsely connected, or missing from the corpus.

**Questions:**

As discussed above in weakness.

---

> ### Author Response · Authors · 2025-11-24
> **Responses to Reviewer jjM6**
>
> We appreciate the reviewer’s valuable feedback and provide our responses to each comment below.
>
> **W1&W2**:The experimental coverage and task diversity are limited. The paper’s practical value for broader complex reasoning tasks remains uncertain.
>
> A: We conduct additional experiments on the Medical datasets from GraphRAG-Bench [1]. These datasets cover several task types, including complex reasoning, contextual summarization, and fact retrieval. We use both LLM-as-a-judge and ROUGE-L to measure performance, following the same evaluation metric as the original paper. As shown in the table, our method consistently outperforms IRCOT across all task categories.
>
> | Task                 | IRCOT LLM_ACC | Ours LLM_ACC | IRCOT ROUGE-L | Ours ROUGE-L |
> |----------------------|----------------|---------------|----------------|---------------|
> | Complex Reasoning    | 0.8389         | **0.8566**        | 0.3577         | **0.3600**        |
> | Contextual Summarize | 0.8166         | **0.8408**        | 0.3569         | **0.3715**        |
> | Fact Retrieval       | 0.8042         | **0.8342**       | 0.3865         | **0.3933**        |
>
> [1] Xiang Z, Wu C, Zhang Q, et al. When to use graphs in rag: A comprehensive analysis for graph retrieval-augmented generation[J]. arXiv preprint arXiv:2506.05690, 2025.
>
> **W3:** The proposed approach underutilizes the structured nature of GraphRAG. Although GraphRAG is designed to leverage entity–relation graphs for structured reasoning, BDTR primarily operates at the text retrieval and reranking level, without explicitly incorporating or reasoning over the underlying graph topology, edge semantics, or relational paths, missing opportunities to exploit the full potential of graph-based representations.
>
> A3: We would like to clarify that our method is fundamentally grounded in the principles of GraphRAG. Although our iterative retrieval module does not operate on an explicitly materialized graph (e.g., an adjacency matrix or graph database), the reasoning stage is graph-guided. The Fast Thought (FT), Slow Thought (ST), and Reasoning Chain (RC) are not free-form textual generations. Instead, the model is prompted to treat the query and retrieved passages as an implicit local graph, and its outputs are constrained by this latent structure. In this sense, FT, ST, and RC function as linguistic interfaces for topological exploration, edge validation, and path construction.
>
> First, the RC requires the model to produce an explicit entity–relation chain, which forces it to extract entities and relations from the retrieved context and commit to a structured multi-hop path. This imposes graph-style supervision rather than allowing unconstrained paraphrasing.
>
> Second, the FT is not a simple reformulation of the original question. It is a targeted query for the next missing node along the chain established in the RC, aligning the model with stepwise graph traversal and path extension rather than purely textual reasoning.
>
> Third, the ST requires the model to justify why a specific bridge entity is needed, making it explicitly articulate how this intermediate node advances the relational path. This justification is inherently topological and operationalizes graph-based neighborhood expansion, even without an explicitly instantiated graph.
>
> Here is our prompt.
> # **prompt**
>
> ---
>
> You are a "Graph Reasoning Agent". Your goal is to navigate a hidden Knowledge Graph to answer complex questions.
> You must identify the "Bridge Entities" that connect the User's Question to the Answer.
>
>
> For every turn, analyze the provided text and produce three specific outputs:
>
> 1. **Reasoning Chain (The Map):**
>    - Construct a valid entity path based on the context.
>    - Format: (Entity A) --[relation]--> (Entity B) --[relation]--> (Target).
>    - CONSTRAINT: Every entity in the chain must be explicitly related to the context. Do not hallucinate links.
>
> 2. **Fast Thought (The Next Step):**
>    - A specific, short question targeting the *next missing node* in your chain.
>    - Example: "Where is [Entity B] located?" (instead of "Where is the answer?")
>
> 3. **Slow Thought (The Topology):**
>    - Explain *why* you are asking the Fast Thought based on graph structure.
>    - Format:
>      **"I have identified [Entity A]. To find [Target], I must first retrieve the neighbor [Entity B] which acts as a bridge."**
>
>
>
> **Rules:**
> - Output format: exactly three lines in this order:
>   1) Reasoning Chain: ...
>   2) Fast Thought: ...
>   3) Slow Thought: ...
>
> ---

---

> ### Author Response · Authors · 2025-11-24
> **Responses to Reviewer jjM6**
>
> **W4:** The paper lacks a detailed analysis of failure cases where the Dual-Thought mechanism fails to retrieve the bridge documents. Without examining such negative examples, it remains unclear how robust the overall retrieval system is under challenging scenarios—e.g., when bridge evidence is implicitly expressed, sparsely connected, or missing from the corpus.
>
> A4: We agree that retrieving bridge documents is crucial, and we explicitly provide both successful and failed cases. As expected, when the bridge evidence is absent from the retrieved set, the model may fail to answer correctly—this is a limitation shared by all retrieval-based methods.
>
> However, our Dual-Thought design specifically aims to mitigate this issue. The two-stage retrieval (Fast Thought + Slow Thought) and the explicit Reasoning Chain together maximize the chance of locating the missing bridge documents. As shown in the successful cases, once the necessary bridge passages are retrieved, our method reliably reconstructs the multi-hop chain and produces the correct answer.
>
> # Successful Case
>
> | Question | What year did the Governor of the city where the basilica named after the same saint as the one that Mantua Cathedral is dedicated to die? |
> |---------|------------------------------------------------------------------------------------------------------------------------------------------------------------------------------------------------------------------------------------------------------------------|
> | Gold Answer | 1952 |
> | Reasoning Chain | (Mantua Cathedral) --[dedicated to]--> (Saint Peter) --[associated city]--> (Vatican City) --[governed by]--> (Governor of Vatican City) --[died in]--> (time) |
> | Prediction Answer | 1952. |
> | Find the bridges | **Bridge 1:** Confirmed that Mantua Cathedral is dedicated to Saint Peter. **Bridge 2:** Identified that St. Peter's Basilica is located in Vatican City. **Bridge 3:** Determined that the Governor of Vatican City (Camillo Serafini) held the post until his death in 1952. |
>
>
>
> # Failure Case
> | Question | An Indy car race was held in the capital of the state where the performer of Mingus Plays Piano was born. Who won the race? |
> |---------|------------------------------------------------------------------------------------------------------------------------------------------------------------------------------------------------------------------------------------------------------------------|
> | Gold Answer | Mario Andretti |
> | RC (Reasoning Chain) | (Charles Mingus) --[born in]--> (Nogales, Arizona) --[race held in]--> (Indianapolis, Indiana) --[winner of]--> (Indy car race in Indianapolis) |
> | Prediction Answer | The winner of the race is not provided in the text. |
> | Missing bridge | The model correctly identified that Charles Mingus was born in Arizona but failed to map the state capital correctly. The capital of Arizona is Phoenix, not Indianapolis, causing the model to search for races in Indianapolis instead of the Phoenix Grand Prix, where Mario Andretti won his final race.

---

### Note · Authors · 2025-12-26

I have read and agree with the venue's withdrawal policy on behalf of myself and my co-authors.